# Local Linear Approximation Algorithm for Neural Network

## Abstract

This paper is concerned with estimation of weights and biases in feed forward neural network (FNN). We propose using local linear approximation (LLA) for the activation function, and develop a LLA algorithm to estimate the weights and biases of one hidden layer FNN by iteratively linear regression. We further propose the layerwise optimized adaptive neural network (LOAN), in which we use the LLA to estimate the weights and biases in the LOAN layer by layer adaptively. We compare the performance of the LOAN with the commonly-used procedures in deep learning via analyses of four benchmark data sets. The numerical comparison implies that the proposed LOAN may outperform the existing procedures.

## 1 Introduction

Although deep learning has many successful applications, the estimation of the weights and biases are still challenging in the construction of deep neural network. This is partly because there is no algorithm to guarantee its resulting solution to be the optimizer of the objective function being optimized. Many heuristics and trial-and-errors methods are applied to get successfully trained network. This work aims to tackle this problem from a different point of view and develop a reliable training algorithm. Consider a $L$-hidden layer FNN:

$$\boldsymbol{h}^{(l)} = \boldsymbol{\sigma}_l(W_l \boldsymbol{h}^{(l-1)} + \boldsymbol{b}_l) \tag{1}$$

for $l = 1, \cdots, L$ with $\boldsymbol{h}^{(0)} = \boldsymbol{x}$, the $p$-dimensional input vector, where $\boldsymbol{\sigma}_l(\cdot)$'s are activation functions, applied to each component of inputs in the previous layer. $W_l$ and $\boldsymbol{b}_l$ are the weight matrix and the bias vector in the $l$-th hidden layer. Consider a linear regression model in the output layer

$$y = \beta_0 + \boldsymbol{\beta}^T \boldsymbol{h}^{(L)} + \varepsilon. \tag{2}$$

Denote $f(\boldsymbol{x}; \boldsymbol{\theta}) = \beta_0 + \boldsymbol{\beta}^T \boldsymbol{h}^{(L)}$, where $\boldsymbol{\theta} = \{\beta_0, \boldsymbol{\beta}, W_1, \cdots, W_L, \boldsymbol{b}_1, \cdots, \boldsymbol{b}_L\}$. Suppose that we have a set of sample $\{\boldsymbol{x}_i, y_i\}$, $i = 1, \cdots, n$. In the literature, the estimation of $\boldsymbol{\theta}$ is to minimize the nonlinear least squares (LS) function

$$\ell(\boldsymbol{\theta}) = \sum_{i=1}^n \{y_i - f(\boldsymbol{x}_i, \boldsymbol{\theta})\}^2,$$

which is highly nonlinear in $\boldsymbol{\theta}$, a high dimensional vector, and hence parameter estimation is challenging for deep neural network. Based on the chain rule, the gradient of $\ell(\boldsymbol{\theta})$ has an expressive form, while the Hessian matrix $\ell''(\boldsymbol{\theta})$ of $\ell(\boldsymbol{\theta})$ is hard to compute, and computational cost for evaluating the inverse of $\ell''(\boldsymbol{\theta})$ is too expensive to be afforded since the dimension of $\boldsymbol{\theta}$ can be huge. As a result, the back propagation with gradient decent or its variations became the most popular parameter estimation method for deep neural network [6, 7, 8, 9]. Also see more recent developments

Submitted to 34th Conference on Neural Information Processing Systems (NeurIPS 2020). Do not distribute.

[1, 10, 12, 13, 17, 18, 11]. We next present a high-level summary of the innovative ideas and major contributions of this work.

We first observe that we could estimate the weights $W_L$ and biases $\boldsymbol{b}_L$ without difficulty if we knew $\boldsymbol{h}^{(L-1)}$. Thus, instead of seeking the optimal weights and biases in all layers simultaneously, we propose to assign the weights and biases layerwise by starting with estimating the weights $W_1$ and $\boldsymbol{b}_1$. To this end, we propose to locally approximate the activation function in FNN by a linear function, and develop a LLA algorithm to estimate $W_l$s and $\boldsymbol{b}_l$s layer by layer based on $\boldsymbol{h}^{(l-1)}$ with $\boldsymbol{h}^{(0)} = \boldsymbol{x}$, the input predictor vector. We refer the neural network constructed by this strategy to as Layerwise Optimized Adaptive neural Network (LOAN for short) by the nature of its construction process. The LLA algorithm is distinguished from existing gradient descent algorithms in that it utilizes the Hessian matrix $\ell''(\boldsymbol{\theta})$ in the same spirit of Fisher scoring algorithm for nonlinear regression models with normal error. By empirical analyses of four benchmark data sets in the literature of deep learning, we compare the performance of the LOAN with the commonly-used deep learning procedures including multi-layer perceptron (MLP) [15], AdaBoost [4], gradient boosting algorithm (GBM) [5], random forest (RF) [2], and XGBoost [3]. Our numerical comparison implies that the one-hidden layer LOAN may outperform these existing procedures in terms of prediction accuracy. Compared with the deep network, the one-hidden layer LOAN enjoys model interpretability and model parsimony. The multiple-hidden layer LOAN can be used to further improve the one-hidden layer LOAN in terms of prediction accuracy.

The rest of this paper is organized as follows. In section 2, we develop the LLA algorithm and an algorithm for construction of the LOAN. Section 3 presents numerical comparisons. Conclusion and discussion are given in section 4.

## 2 New method for constructing neural network

Let us start with one-hidden layer neural network in order to get insights into the LLA algorithm.

### 2.1 One-hidden layer neural network

Note that $\boldsymbol{h}^{(0)} = \boldsymbol{x}$. Then one hidden-layer FNN with $J_1$ nodes can be expressed as follows $\boldsymbol{h}^{(1)} = \boldsymbol{\sigma}(W\boldsymbol{h}^{(0)} + \boldsymbol{b}) = (\sigma(\boldsymbol{w}_1^T \boldsymbol{x} + b_1), \cdots, \sigma(\boldsymbol{w}_{J_1}^T \boldsymbol{x} + b_{J_1}))^T$, where we drop the subscripts of $W$ and $\boldsymbol{b}$ for ease of presentation. Throughout this paper, we set $\sigma(z) = \max\{z, 0\}$, the ReLU activation function. To estimate the weights and biases, we minimize a nonlinear LS function

$$\ell_1(\boldsymbol{\theta}) = \sum_{i=1}^n \{y_i - \beta_0 - \sum_{j=1}^{J_1} \beta_j \sigma(\boldsymbol{w}_j^T \boldsymbol{x}_i + b_j)\}^2. \tag{3}$$

We next propose an algorithm to minimize (3). Given $\boldsymbol{w}_j^{(c)}$ and $b_j^{(c)}$ in the current step, we propose to approximate $\sigma(\boldsymbol{x}^T \boldsymbol{w}_j + b_j)$ by a linear function based on the first-order Taylor expansion of $\sigma(z)$:

$$\sigma(\boldsymbol{x}^T \boldsymbol{w}_j + b_j) \approx \sigma(\boldsymbol{x}^T \boldsymbol{w}_j^{(c)} + b_j^{(c)}) + \{(\boldsymbol{x}^T \boldsymbol{w}_j + b_j) - (\boldsymbol{x}^T \boldsymbol{w}_j^{(c)} + b_j^{(c)})\} I(\boldsymbol{x}^T \boldsymbol{w}_j^{(c)} + b_j^{(c)} > 0) \tag{4}$$

for $j = 1, \cdots, J_1$. We refer (4) to as local linear approximation (LLA). Thus,

$$\beta_j \sigma(\boldsymbol{x}^T \boldsymbol{w}_j + b_j) \approx \beta_j \sigma(\boldsymbol{x}^T \boldsymbol{w}_j^{(c)} + b_j^{(c)}) + \gamma_j I(\boldsymbol{x}^T \boldsymbol{w}_j^{(c)} + b_j^{(c)} > 0) + \boldsymbol{\eta}_j^T \boldsymbol{x} I(\boldsymbol{x}^T \boldsymbol{w}_j^{(c)} + b_j^{(c)} > 0) \tag{5}$$

where $\gamma_j = \beta_j(b_j - b_j^{(c)})$ and $\boldsymbol{\eta}_j = \beta_j(\boldsymbol{w}_j - \boldsymbol{w}_j^{(c)})$. Define $z_{1ij} = \sigma(\boldsymbol{x}_i^T \boldsymbol{w}_j^{(c)} + b_j^{(c)})$, $z_{2ij} = I(\boldsymbol{x}_i^T \boldsymbol{w}_j^{(c)} + b_j^{(c)} > 0)$, and $\boldsymbol{z}_{3ij} = \boldsymbol{x}_i I(\boldsymbol{x}_i^T \boldsymbol{w}_j^{(c)} + b_j^{(c)} > 0)\}$, $j = 1 \cdots, J_1$. Further define $\boldsymbol{z}_{1i} = [z_{1i1}, \cdots, z_{1iJ_1}]$, $\boldsymbol{z}_{2i} = [z_{2i1}, \cdots, z_{2iJ_1}]$, $\boldsymbol{z}_{3i} = [\boldsymbol{z}_{3i1}^T, \cdots, \boldsymbol{z}_{3iJ_1}^T]$, and $\boldsymbol{z}_i = [\boldsymbol{z}_{1i}, \boldsymbol{z}_{2i}, \boldsymbol{z}_{3i}]^T$, which is a $J_1(p+2)$-dimensional vector. With the aid of approximation (5), the objective function in (3) is approximated by

$$\sum_{i=1}^n \{y_i - \beta_0 - \sum_{j=1}^{J_1} \{\beta_j z_{1ij} + \gamma_j z_{2ij} + \boldsymbol{\eta}_j^T \boldsymbol{z}_{3ij}\}\}^2, \tag{6}$$

which is the LS function of linear regression with the response $y_i$ and predictors $\boldsymbol{z}_i$. Denote the resulting LS estimate of $\beta_j$, $\gamma_j$ and $\boldsymbol{\eta}_j$ by $\hat{\beta}_j$, $\hat{\gamma}_j$ and $\hat{\boldsymbol{\eta}}_j$, respectively. By the definition of $\gamma_j$ and $\boldsymbol{\eta}_j$,

we can update $b_j$ and $\boldsymbol{w}_j$ as follows

$$b_j^{(c+1)} = b_j^{(c)} + \hat{\gamma}_j/\hat{\beta}_j, \text{ and } \boldsymbol{w}_j^{(c+1)} = \boldsymbol{w}_j^{(c)} + \hat{\boldsymbol{\eta}}_j/\hat{\beta}_j. \tag{7}$$

If $|\hat{\beta}_j|$ is very close to zero, one may simply set $b_j^{(c+1)} = b_j^{(c)}$ and $\boldsymbol{w}_j^{(c+1)} = \boldsymbol{w}_j^{(c)}$. Thus, we may estimate $W$ and $\boldsymbol{b}$ by iteratively updating (7) and regressing $y_i$ on the updated $\boldsymbol{z}_i$. The procedure can be summarized as the following algorithm.

**Algorithm 1: Local Linear Approximation (LLA) Algorithm**

**Step 1** Set initial value for $W^{(0)} = [\boldsymbol{w}_1^{(0)}, \cdots, \boldsymbol{w}_{J_1}^{(0)}]^T$ and $b_j^{(0)}$, and let $c = 0$.

**Step 2** Calculate $\boldsymbol{z}_i$ defined in the text based on $\boldsymbol{w}_j^{(c)}$ and $b_j^{(c)}$, obtain the least squares estimate (LSE) $\hat{\beta}_j$s, $\hat{\gamma}_j$s and $\hat{\boldsymbol{\eta}}_j$s by running a linear regression $y_i$ on covariate $\boldsymbol{z}_i$, and update the biases and weights

$$b_j^{(c+1)} = b_j^{(c)} + \hat{\gamma}_j/\hat{\beta}_j, \text{ and } \boldsymbol{w}_j^{(c+1)} = \boldsymbol{w}_j^{(c)} + \hat{\boldsymbol{\eta}}_j/\hat{\beta}_j.$$

Set $c = 1, 2, \cdots$, and repeat Step 2 until the criterion of algorithm convergence meets.

We propose how to construct initial values for the iterative updates in section 2.3

## 2.2 Layer-wise Optimized Adaptive Neural Network

The back-propagation with gradient decent algorithm is to seek optimal weights and biases in all layers of FNN via minimizing $\ell(\boldsymbol{\theta})$ in (2). This leads to a high-dimensional, nonconvex minimization problem. We propose a new procedure to assign the weights and biases as follows. We start with $\boldsymbol{h}^{(0)} = \boldsymbol{x}$, fit the data with one-hidden layer network as described in section 2.1, and obtain an estimate $\hat{W}$ and $\hat{\boldsymbol{b}}$ of $W$ and $\boldsymbol{b}$. We set $W_1 = \hat{W}$ and $\boldsymbol{b}_1 = \hat{\boldsymbol{b}}$ for the weight matrix and bias vector in FNN. Then we define $\tilde{\boldsymbol{x}}_i = \boldsymbol{h}_i^{(1)} = \boldsymbol{\sigma}(W_1 \boldsymbol{h}_i^{(0)} + \boldsymbol{b}_1)$. We fit data $\{\tilde{\boldsymbol{x}}_i, y_i\}$ with one-hidden layer network, and obtain $\hat{W}$ and $\hat{\boldsymbol{b}}$ by the proposed LLA. Then we set $W_2 = \hat{W}$ and $\boldsymbol{b}_2 = \hat{\boldsymbol{b}}$. Thus, we construct a LOAN by estimating $W_l$'s and $\boldsymbol{b}_l$'s by running one-hidden layer network on data $\{\tilde{\boldsymbol{x}}_i, y_i\}$ with $\tilde{\boldsymbol{x}}_i = \boldsymbol{h}_i^{(l-1)}$. This procedure can be summarized as the following algorithm.

## Algorithm 2: Layerwise Optimized Adaptive Network (LOAN)

**Step 1** Input LOAN structure $[J_1, \cdots, J_L]$ with $J_l$ being the number of nodes for the $l$-th layer.

**Step 2** Obtain $W_1$ and $\boldsymbol{b}_1$ in the 1st hidden layer by fitting the data $\{\boldsymbol{x}_i, y_i\}$ to a one-hidden layer neural network by the LLA algorithm. Set $\boldsymbol{h}_i^{(1)} = \boldsymbol{\sigma}(W_1 \boldsymbol{x}_i + \boldsymbol{b}_1)$.

**Step 3** For $l = 1, \cdots, L-1$, set $\tilde{\boldsymbol{x}}_i = \boldsymbol{h}^{(l)}$, and obtain $W_{l+1}$ and $\boldsymbol{b}_{l+1}$ by fitting the data $\{\tilde{\boldsymbol{x}}_i, y_i\}$ to one-hidden layer neural network by the LLA algorithm. Set $\boldsymbol{h}_i^{(l+1)} = \boldsymbol{\sigma}(W_{l+1} \tilde{\boldsymbol{x}}_i + \boldsymbol{b}_{l+1})$.

## 2.3 Initial values for the LLA

Motivated by model fitting for a single index model, we propose an initial value for using the LLA algorithm to estimate $W_l$ and $\boldsymbol{b}_l$ at the $l$-th layer with $J_l$ nodes as follows. Set $\tilde{\boldsymbol{x}} = \boldsymbol{h}^{(l-1)}$, we fit data $\{\tilde{\boldsymbol{x}}_i, y_i\}$ to a linear regression model and obtain the LS estimate of coefficients of $\tilde{\boldsymbol{x}}$, denoted by $\hat{\boldsymbol{\alpha}}$. Let $\hat{\tau}_j$, $j = 1, \cdots, J_l$ be quantile points of $\tilde{\boldsymbol{x}}_i^T \hat{\boldsymbol{\alpha}}$, $i = 1, \cdots, n$. We may set initial value of $b_j^{(0)} = -\hat{\tau}_j$, and $\boldsymbol{w}_j^{(0)} = \hat{\boldsymbol{\alpha}}$ for $j = 1, \cdots, J_l$.

Since the objective function $\ell_1(\boldsymbol{\theta})$ is nonconvex, numerical minimization algorithm may depend on the initial value. We find the proposed initial value strategy in the last paragraph work reasonably well in practice, but the performance of the LOAN can be further improved by integrating model averaging and cross-validation strategy. Specifically, We randomly partition data into a pre-specified $K$ subsets with approximately the same sample size, and then for $k = 1, \cdots, K$, we fit a LOAN with data excluding the $k$-th data subset, and obtain the mean squared error (MSE) for the entire data set, denoted by $\text{MSE}_k$. Rank the resulting LOANs by their MSEs from smallest to largest. Set the final model to be the average of the top $K_s$ LOANs with smaller MSEs. Here $K_s \leq K$ is a pre-specified integer. In our numerical study, we set $K = 20$ and $K_s = 15$, and find this work well.

Table 1: Comparison with commonly-used procedures with default hyper-parameter value.

| Methods | AF
100*mean(std) | BS
100*mean(std) | CHP
100*mean(std) | PK
100*mean(std) |
|---|---|---|---|---|
| LOAN | 4.3317(0.5895) | 7.8386(0.4148) | 18.5801(0.6955) | 42.6082(4.9884) |
| MLP | 25.4913(3.4600) | 9.9173(2.0827) | 24.1495(1.0754) | 31.4595(2.3832) |
| XGBoost | 5.1042(0.7245) | 7.8751(0.3137) | 16.4491(0.6905) | 3.7880(0.4386) |
| RF | 6.5254(0.7346) | 10.2843(0.4672) | 21.2067(0.9217) | 2.8041(0.4835) |

Table 2: Comparison of the impact of PCA on LOAN and existing procedures.

| Methods | 6 PCs
100*mean(std) | 9 PCs
100*mean(std) | 13 PCs
100*mean(std) | All (19) PCs
100*mean(std) |
|---|---|---|---|---|
| LOAN | 29.3152(2.7984) | 18.3863(2.8369) | 27.2601(7.1044) | 43.0542(3.6137) |
| MLP | 41.7293(2.6673) | 34.7943(2.3511) | 30.6869(2.1018) | 30.6617(2.9831) |
| XGBoost | 50.8823(3.2462) | 47.1367(2.7843) | 42.9631(2.6408) | 44.4751(2.1000) |
| RF | 41.2433(2.8571) | 39.6862(2.6231) | 37.7282(2.5413) | 39.9511(2.5392) |

## 3  Numerical comparison

This section provides a brief summary of our numerical comparison. A complete and detailed description of data sets including the sample size, dimension of predictors, and the version of existing procedures are given in the Appendix of this paper. In this section, we compare the performance of the proposed LOAN with MLP [15], AdaBoost [4], GBM [5], RF [2], and XGBoost [3] by empirical analyses of four benchmark data sets: airfoil (AF, for short) data, bikesharing (BS, for short) data, california house price (CHP, for short) data, and parkinson (PK, for short) data. XGBoost performs better than AdaBoost and GBM. Thus, we present results of XGBoost only to save space. Results of AdaBoost and GBM are given in the Appendix.

In our numerical comparison, we first standardize the response and predictors so that their sample means and variances equal 0 and 1, respectively. We split each data set into 80% training data and 20% testing data with seed $1, \cdots, 30$. Thus, we may obtain 30 MSEs for testing data and 30 MSEs for training data for each procedure. We report the sample mean and standard deviation (std) of the 30 MSEs for each procedure and data set. We first compare the performance of the one-hidden layer LOAN with 60 nodes with all other procedures with default settings. Table 1 depicts the values of 100 times of the sample mean and std of the 30 MSEs, and implies that the LOAN performs well in terms of prediction accuracy. Specifically the LOAN has the smallest MSE for AF and BS data sets. For CHD, XGboost performs the best, and follows by the LOAN. The LOAN has slightly greater MSE than XGBoost since XGBoost has a deeper tree by default. For PK data, the RF performs the best, and then follows by XGBoost. It seems that both RF and XGBoost perform much better than the LOAN and MLP. This motivates us to examine the data further. By doing exploratory data analysis of PK data, we find that several of its 19 predictors are highly correlated. The poor performance of the LOAN may be due to the collinearity of predictors since the linear regression is used to update the weights and biases. We conduct a principal component analysis (PCA), and the first 6, 9 and 13 principal components (PC) can explain 95%, 99% and 99.9% of the total variance. We further apply all procedures to PK data with predictors being set to PCs. Table 2 depicts the MSEs for PK data with PC predictors. Compared with Table 1, the performance of XGBoost and RF becomes poor. The performance of MLP is quite stable across different numbers of PCs. The performance of the LOAN improves significantly via using PCs. In Tables 1 and 2, we use the one-layer LOAN for comparison. We have examine the performance of the multiple-hidden layer LOAN in the Appendix due to the space limit.

## 4  Conclusion and discussion

We develop the LLA algorithm and further develop the LOAN. Our comparison indicates that the LOAN may outperform commonly-used procedures. This paper focuses on regression with continuous responses. The LOAN can be extended to classification problems.

## Broader Impact

FNN is a commonly-used model in deep learning, which has been used in our daily life. The proposed LLA algorithm can significantly improve the existing algorithm for FNN, and the proposed LOAN can outperform commonly-used procedures such as MLP and XGBoost in deep learning. Researchers in data mining and machine learning can certainly benefit from the proposed LLA algorithm and LOAN. People who use deep learning techniques and methods may also benefit this research in their daily life.

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

**Appendix of "Local Linear Approximation Algorithm for Neural Network"**

193  This appendix provides a complete section of Section 3 on numerical comparison in the main text. We
194  compare the performance of the proposed LOAN with commonly-used procedures in deep learning
195  by empirical analysis of four data sets:

196  (1) airfoil (AF, for short) data, which consists of $n = 1503$ observations with $p = 5$ predictors and
197      can be downloaded from
198      `https://archive.ics.uci.edu/ml/datasets/Airfoil+Self-Noise`,
199      The response is the scaled sound pressure level. The explanatory variables include frequency,
200      angle of attack, chord length, free-stream velocity, suction side displacement thickness

201  (2) bikesharing (BS, for short) data, which consists of $n = 17379$ observations with $p = 8$ predictors
202      and can be downloaded from
203      `https://archive.ics.uci.edu/ml/datasets/bike+sharing+dataset`.
204      The response is the log-transformed count. The explanatory variables include normalized
205      hour, normalized temperature, normalized humidity, normalized wind speed, season, holiday,
206      weekday and weather situation.

207  (3) California house price (CHP, for short) data [14], which consists of $n = 20,640$ observations
208      with $p = 8$ predictors and can be downloaded from Carnegie-Mellon *StatLib* repository
209      (http://lib.stat.cmu.edu/datasets/).
210      The response variable is the median house value. The explanatory variables include longi-
211      tude, latitude, housing median age, medium income, population, total rooms, total bedrooms
212      and households.

213  (4) Parkinson (PK, for short) data, which consists of $n = 5874$ observations with $p = 19$ predictors
214      and can be downloaded from
215      `https://archive.ics.uci.edu/ml/datasets/Parkinsons+Telemonitoring`.
216      The response variable is the total UPDRS scores, and the explanatory variables include
217      subject age, subject gender, time interval from baseline recruitment date, and 16 biomedical
218      voice measures.

219  In our numerical anlaysis, we compare the performance of the proposed LOAN with

220  1. MLP: multi-layer perceptron [15, 16] (python scikit-learn version 0.22.1),

221  2. XGBoost [3] (python version 1.2.0).

222  3. GBM: gradient boosting algorithm [5, 16]

223  4. AdaBoost [4, 16]

224  5. RF: random forest [2, 16]

225  In our numerical comparison, we first standardize the response and predictors so that their sample
226  means and variances equal 0 and 1, respectively. We split each data set into 80% training data and
227  20% testing data with seed $1, \cdots, 30$. Thus, we may obtain 30 MSEs for testing data and 30 MSEs
228  for training data for each procedure. We report the sample mean and standard deviation (std) of the
229  30 MSEs for each procedure and data set.

230  We first compare the performance of the one-hidden layer LOAN with all other procedures with
231  default settings. The number of nodes in the LOAN is set to be 60. Table A.1 depicts the values of
232  100 times of the sample mean and std of the 30 MSEs of testing data (testing MSE for short) as well
233  as the 30 MSEs of training data (training MSE for short). Table A.1 implies that XGBoost performs
234  better than other two boosting methods: AdaBoost and GBM, and the LOAN performs well in terms
235  of prediction accuracy. Specifically the LOAN has the smallest testing MSEs for AF and BS data sets.
236  For CHD, XGboost performs the best, and follows by the LOAN in terms of testing MSE. The LOAN
237  has slightly greater testing MSE than XGBoost, while XGBoost has a deeper tree by default. For
238  AF, BS and CHD data sets, the LOAN has the smallest difference between testing MSE and training
239  MSE. This implies that the LOAN is less likely to be over-fitting. Compared with other procedures,
240  the LOAN has relatively small standard deviation of MSE. This implies that the LOAN has a fairly
241  stable performance. For PK data, the RF performs the best, and then follows by XGBoost. It seems
242  that both RF and XGBoost performs much better than the LOAN and MLP. This motivates us to
243  examine the data further.

Table A.1: Comparison with commonly-used procedures with default hyper-parameter value.

| Methods | AF 100*mean(std) | BS 100*mean(std) | CHP 100*mean(std) | PK 100*mean(std) |
|---|---|---|---|---|
| | | MSE for Testing Data | | |
| LOAN | 4.3317(0.5895) | 7.8386(0.4148) | 18.5801(0.6955) | 42.6082(4.9884) |
| MLP | 25.4913(3.4600) | 9.9173(2.0827) | 24.1495(1.0754) | 31.4595(2.3832) |
| XGBoost | 5.1042(0.7245) | 7.8751(0.3137) | 16.4491(0.6905) | 3.7880(0.4386) |
| AdaBoost | 31.7242(2.6197) | 27.7419(0.7384) | 56.3744(5.3267) | 60.9171(1.8358) |
| GBM | 14.7338(1.5846) | 16.5968(0.5592) | 21.2866(0.7254) | 21.0340(0.7685) |
| RF | 6.5254(0.7346) | 10.2843(0.4672) | 21.2067(0.9217) | 2.8041(0.4835) |
| | | MSE for Training Data | | |
| LOAN | 2.3213(0.1634) | 6.3305(0.2549) | 15.7216(0.1275) | 28.8462(2.5983) |
| MLP | 24.5284(0.0268) | 7.7488(0.7496) | 23.7055(1.1570) | 23.7447(1.3502) |
| XGBoost | 0.2623(0.0268) | 4.3713(0.0946) | 5.7634(0.1456) | 0.3115(0.0264) |
| AdaBoost | 29.2881(1.1124) | 27.2715(0.6059) | 55.8631(5.5463) | 60.4119(1.1948) |
| GBM | 11.2280(0.6303) | 15.8681(0.1972) | 19.5493(0.1864) | 19.0881(0.7516) |
| RF | 0.9366(0.0388) | 1.4413(0.0179) | 3.8432(0.0766) | 0.3897(0.0356) |

Table A.2: Comparison of the impact of PCA on LOAN and existing procedures based on PK data

| Methods | 6 PCs 100*mean(std) | 9 PCs 100*mean(std) | 13 PCs 100*mean(std) | All (19) PCs 100*mean(std) |
|---|---|---|---|---|
| | | MSE for Testing Data | | |
| LOAN | 29.3152(2.7984) | 18.3863(2.8369) | 27.2601(7.1044) | 43.0542(3.6137) |
| MLP | 41.7293(2.6673) | 34.7943(2.3511) | 30.6869(2.1018) | 30.6617(2.9831) |
| XGBoost | 50.8823(3.2462) | 47.1367(2.7843) | 42.9631(2.6408) | 44.4751(2.1000) |
| AdaBoost | 81.2930(2.6109) | 78.9303(2.7339) | 76.2363(2.2727) | 76.5351(2.5682) |
| GBM | 67.0713(2.3618) | 62.4927(2.2980) | 59.1466(1.8239) | 59.7928(1.9610) |
| RF | 41.2433(2.8571) | 39.6862(2.6231) | 37.7282(2.5413) | 39.9511(2.5392) |
| | | MSE for Training Data | | |
| LOAN | 21.7594(1.5550) | 11.8719(1.4716) | 16.0485(2.7811) | 29.2183(3.0557) |
| MLP | 37.1915(1.8439) | 27.9013(0.9417) | 22.0252(0.9036) | 20.5078(0.9063) |
| XGBoost | 10.0009(0.7694) | 6.6056(0.4240) | 4.0770(0.3162) | 2.9568(0.2952) |
| AdaBoost | 79.1353(2.3975) | 76.3020(2.1195) | 73.5840(2.0781) | 73.5073(2.3712) |
| GBM | 59.1133(1.5346) | 53.7506(1.1314) | 49.5714(0.9834) | 49.4500(0.9870) |
| RF | 5.9341(0.2602) | 5.6795(0.1998) | 5.3804(0.2432) | 5.6393(0.2185) |

By doing exploratory data analysis of PK data, we find that several of the 19 predictors in PK data are highly correlated. For example, the correlation between Shimmer and Shimmer(dB) is 0.9923. The correlation between Jitter($\%$) and Jitter(RAP) is 0.9841. The poor performance of the LOAN may be due to the collinearity of predictors since the linear regression is applied for updating the weights and biases. Thus, we conduct a principal component analysis (PCA), and the first 6, 9 and 13 principal components (PC) can explain 95%, 99% and 99.9% of the variance. We further apply all procedures to PK data with predictors being set to PCs. Table A.2 depicts the MSEs for PK data with PC predictors. Compared with Table A.1, the performance of XGBoost and RF becomes poor. Tables A.1 and A.2 clearly imply that the performance of RF and XGBoost are not robust under linear transformation on predictors. The performance of MLP is quite stable across different numbers of PC variables. The performance of the LOAN improves significantly via using PCs.

In Tables A.1 and A.2, we use the one hidden-layer LOAN for comparison. It is of interest to examine the performance of the LOAN with multiple-hidden layers. To this end, we set the MLP, XGBoost, GBM and RF with the same number of layer or depth as $L$, the number of hidden layers used in the LOAN. Other parameters of the MLP, XGBoost, GBM and RF are set to be the default values, We apply all these procedures to CHD data. Table A.3 depicts the sample means and standard deviation

Table A.3: Comparison of the number of hidden layers in LOAN and existing procedures based on CHD data

| Methods | $L = 1$ 100*mean(std) | $L = 2$ 100*mean(std) | $L = 3$ 100*mean(std) |
|---------|------------------------|------------------------|------------------------|
| | MSE for Testing Data | | |
| LOAN | 18.5801(0.6955) | 18.0160(0.6742) | 17.5237(0.6530) |
| MLP | 22.5382(0.9667) | 20.6272(0.8958) | 20.2324(0.8761) |
| XGBoost | 28.5367(0.7766) | 20.7961(0.6612) | 18.3401(0.6515) |
| GBM | 35.5315(0.8693) | 24.5913(0.7452) | 21.2869(0.7255) |
| RF | 66.7451(1.4958) | 53.6974(1.1509) | 43.9402(1.0961) |
| | MSE for Training Data | | |
| LOAN | 15.7216(0.1275) | 14.3645(0.2425) | 12.5553(0.1572) |
| MLP | 21.2712(0.6557) | 18.3534(0.4721) | 16.7369(0.5431) |
| XGBoost | 27.9085(0.1824) | 19.1496(0.2419) | 15.1269(0.1666) |
| GBM | 35.1605(0.1962) | 23.7616(0.2104) | 19.5492(0.1863) |
| RF | 66.7071(0.6086) | 53.5375(0.3329) | 43.4837(0.3294) |

of the 30 training and testing data splittings for $L = 1, 2$ and 3. The number of nodes in the LOAN with one hidden layer is set to be 60. The number of nodes in the LOAN with two hidden layer is set to be 60 and 10 in the first and second layer, respectively. The number of nodes in the LOAN with three hidden layers is set to be 60, 10 and 50 in the first, second and third layer, respectively. Since there is no option to set different depth levels for AdaBoost, we exclude it for this comparison. From Table A.3, we can see that both the LOAN and MLP have smaller MSE as $L$ increases, but there is no dramatic improvement, while XGBoost, GBM and RF seem to have more improvement as $L$ increases. Table A.3 indicates that the LOAN performs the best when $L = 1, 2$ and 3.

