# OpenReview forum: "Local Linear Approximation Algorithm for Neural Network"
_NeurIPS.cc/2020/Workshop/DL-IG — Submitted to NeurIPSW 2020: DL-IG_

### Official Review · AnonReviewer2 · 2020-10-29
**Review of "Local Linear Approximation Algorithm for Neural Network"**

**Rating:** 5
**Confidence:** 4

**Review:**

This paper considers a method to locally learn the weights for neural networks without backpropagation by locally solving an optimization problem at each layer. Although I'm not very familiar with this line of work, I have run across many related ideas which were not mentioned in this paper. I list them below to help put the ideas in this paper in better context in future iterations. Also, I don't think that this workshop is a particularly relevant venue for this work.

Related work
I'd look into three different lines of work that are relevant to what you've described. I've just given one or two example papers, hopefully the related work will help expand into more relevant directions.

- Synthetic gradients. This has a similar flavor - you locally solve an optimization problem at each layer.
Jaderberg et al. "Decoupled Neural Interfaces using Synthetic Gradients"

- The credit assignment problem. This line of work discusses the heart of the problem: how do we figure out how a weight in an early layer is responsible for getting a correct or incorrect prediction?
Lansdell et al, ICLR 2020. "Learning to Solve the Credit Assignment Problem".

- Biologically plausible learning without backpropagation. A lot of work has been motivated by the idea that backprop seems unlikely to occur in the brain, but that the local dynamics in the brain somehow manage to learn well anyway.
Bengio et al. Towards Biologically Plausible Deep Learning
Lowe et al, NeurIPS 2019. Putting An End to End-to-End: Gradient-Isolated Learning of Representations
It's not directly related, but based on the technique you used you might also enjoy Song et al, "Nonlinear Equation Solving: A Faster Alternative to Feedforward Computation".

---

### Official Review · AnonReviewer1 · 2020-11-07

**Rating:** 4
**Confidence:** 4

**Review:**

This paper develops a linear approximation to a neural network. The method involves fitting the weights and biases of a network for a regression task sequentially starting from the first layer to the final one. Within each layer, the authors use a linear approximation of the nonlinearity. Empirical results are shown on some classical regression datasets.

I think this submission is very premature and not a good fit for this workshop. A linear approximation of a neural network will perform poorly for difficult datasets. The experimental results in Table 1 are difficult to believe, how can an MLP (which should perform at least as good as the greedy training procedure) have such a a large error? I would encourage the authors to take a careful look at existing literature to refine this idea. In particular, the greedy training reminds one of contrastive divergence used for training Boltzmann machines.

---

### Decision · Program_Chairs · 2020-11-07

**Decision:**

Reject

**Comment:**

Dear authors, both reviewers point out significant deficiencies in the current manuscript and feel that this paper is not a good fit for the present workshop. You are encouraged to use this feedback to improve upon this idea further.